# Fibre Intake Is Associated with Cardiovascular Health in European Children

**DOI:** 10.3390/nu13010012

**Published:** 2020-12-23

**Authors:** Susana Larrosa, Veronica Luque, Veit Grote, Ricardo Closa-Monasterolo, Natalia Ferré, Berthold Koletzko, Elvira Verduci, Dariusz Gruszfeld, Annick Xhonneux, Joaquin Escribano

**Affiliations:** 1Paediatrics, Nutrition and Development Research Unit, Universitat Rovira i Virgili, IISPV, 43204 Reus, Spain; slarrosa@grupsagessa.com (S.L.); ricardo.closa@urv.cat (R.C.-M.); natalia.ferre@urv.cat (N.F.); 2Serra Hunter Fellow, Universitat Rovira i Virgili, 43201 Reus, Spain; 3Department Paediatrics, Dr. von Hauner Children’s Hospital, University Hospital, LMU Ludwig-Maximilians-Universität, 80337 Munich, Germany; veit.grote@med.uni-muenchen.de (V.G.); berthold.koletzko@med.uni-muenchen.de (B.K.); 4Else Kröner-Seniorprofessor of Paediatrics, LMU Ludwig-Maximilians-Universität, 80337 Munich, Germany; 5Department of Health Sciences, University of Milan, 20146 Milano, Italy; elvira.verduci@unimi.it; 6Department of Pediatrics Ospedale Vittore Buzzi, University of Milan, 20154 Milano, Italy; 7Neonatal Department, Children’s Memorial Health Institute, 04-730 Warsaw, Poland; d.gruszfeld@ipczd.pl; 8CHC Sant Vincent, 4000 Liège-Rocourt, Belgium; annick.xhonneux@chc.be

**Keywords:** dietary fibre, children, cardiovascular risk, soluble fibre, insoluble fibre, resistant starch, fibre food sources

## Abstract

Background: We aimed at analysing the association between dietary fibre intake during childhood and cardiovascular health markers. Methods: We used observational longitudinal analysis and recorded diet using 3-day diaries at the ages of 3, 4, 5, 6, and 8 years in children from the EU Childhood Obesity Project Trial. At the age of 8, waist circumference, systolic and diastolic blood pressure (SBP and DBP) and biochemical analyses (lipoproteins, triglycerides and homeostasis model for insulin resistance (HOMA-IR)) were evaluated. Those parameters were combined into a cardiometabolic risk score through the sum of their internal z-scores. Results: Four-hundred children (51.8% girls) attended to the 8-year visit with a 3-day diary. Adjusted linear regression models showed that children who repeatedly stayed in the lowest tertile of fibre intake during childhood had higher HOMA-IR (*p* = 0.004), higher cardiometabolic risk score (*p* = 0.02) and a nonsignificant trend toward a higher SBP at 8 years. The higher the dietary intake of soluble fibre (from fruits and vegetables) at 8 years, the lower the HOMA-IR and the cardiometabolic risk score (*p* = 0.002; *p* = 0.004). SBP was directly associated with fibre from potatoes and inversely with fibre from nuts and pulses. Conclusion: A diet rich in dietary fibre from fruits, vegetables, pulses and nuts from early childhood was associated to a healthier cardiovascular profile, regardless of children’s weight.

## 1. Introduction

An optimal diet during childhood must be adequate to support normal growth and development. At the same time, the diet must be aimed at reducing the risk of diet-related chronic diseases during adulthood. Starting proper dietary habits from early childhood could help to maintain them throughout life [1]. Dietary fibre intake is frequently related to the digestive and cardiovascular health of the adult and child population [2,3]. Physiological benefits of fibre on health are produced by several of its properties, such as viscosity, solubility and fermentability. The main effects derived from the viscosity of soluble fibre are responsible for reducing lipid absorption, slowing carbohydrate absorption, producing satiety and providing part of its anticarcinogenic potential. Insoluble fibres produce an increase in faecal mass that accelerates intestinal transit, which is useful in the treatment and prevention of chronic constipation and colon cancer [4,5]. More recent studies have highlighted the fermentability of the fibre, in contact with the colonic microflora, as the most important property, since a multitude of local and systemic effects derive from it [6]. The short-chain fatty acids, which result from the fermentation process [7], are responsible for several functions related to the cardiovascular and digestive tract health, such as an anti-inflammatory action, protection against colonic carcinogenesis, a direct effect on the synthesis of cholesterol and glucose, a reduction of peripheral insulin resistance and a prebiotic effect [8,9,10,11,12,13,14,15].

In adults, a low dietary fibre intake has been related to risk indicators of cardiovascular disease such as hypercholesterolemia, diabetes mellitus, high blood pressure and obesity, as well as gastrointestinal diseases including constipation, irritable bowel syndrome, ulcerative colitis, diverticular disease and colorectal cancer [16,17,18,19]. Currently, it is proposed that a dietary fibre intake between 25–35 g a day in adults may contribute to reducing the prevalence of some of these noncommunicable diseases [20,21,22].

Studies in children analysing the association between dietary fibre intake and cardiovascular health are scarce and have been performed mainly in adolescence [23,24]. Therefore, studies are needed to evaluate the relationship of dietary fibre intake with health indicators in young children. We aimed at analysing the relation of the dietary fibre intake in children from five European countries at the ages of 3 to 8 years to cardiovascular health markers such as obesity, blood lipids, blood pressure and glucose metabolism.

## 2. Materials and Methods

### 2.1. Study Design and Population

This was an observational longitudinal analysis assessing the association between dietary fibre intakes during childhood, from 3 to 8 years, on cardiovascular health at 8 years. This study is a secondary analysis of data collected in the EU-CHOP Childhood Obesity Project trial (NCT00338689), carried out in Germany, Belgium, Italy, Poland and Spain. The EU-CHOP was a double-blind, randomized dietary intervention trial that recruited children from birth (October 2002 to July 2004) and followed their dietary intake until the age of 8 years. Recruited infants were healthy, full-term and born from uncomplicated pregnancies with a normal weight for gestational age. Formula-fed infants were randomly assigned to either a lower or a higher protein content formula, with the aim to assess the effect on later obesity risk. A group of breastfed infants was recruited and followed up as the observational gold standard group. Details of the clinical trial have been previously published [25,26]. The data used for this study were collected prospectively during the post-intervention follow-up visits.

### 2.2. Dietary Intake Assessment

Dietary intake was recorded using 3-day diet diaries at the ages of 3, 4, 5, 6, and 8 years. The amounts of recorded foods were coded by trained research nutritionists following standardized procedures [27,28] and were introduced in a dedicated program for the conversion to nutrients. The program contained the German BLS II.2 food composition database [29], and the nutrient content of local foods from the different countries was added using national food composition databases. The average intakes of energy (kcal/day), fibre (g/day, g/1000 kcal) and macronutrients (g/day, g/1000 kcal and as % of total energy) of the 3 days were calculated. Furthermore, the amount of fibre (g, g/1000 kcal) was quantified according to its food source as that from cereals and derivatives (as an approximation to insoluble fibre), fruits and vegetables (as an approximation to soluble fibre), potatoes and tubers (as an approximation to resistant starch) and that from legumes and nuts that contain different types of soluble and insoluble fibres and resistant starch [30].

### 2.3. Health Variables

At the age of 8, we measured anthropometry. The main health outcome measures were blood pressure and blood sample parameters related to cardiovascular risk.

#### 2.3.1. Anthropometry

At 8 years of age, the measurements of weight (kg), height (cm), calculation of the body mass index (BMI) (kg/m^2^) and waist circumference (cm) were analysed. Weight was measured in underwear on a SECA 702/703 digital scale (10 g precision). Height was measured with a SECA 242 digital statiometer (1 mm precision). Standardized procedures for measuring the child’s height included having the feet slightly apart, with the back of the head, shoulder blades, buttocks and heels touching the vertical board whenever possible. The mother or caregiver was asked to hold the child’s knees and ankles to help keep the legs straight while the researcher held the child’s head in the Frankfort plane and read the measurement. Weight, height and BMI z-scores were calculated using World Health Organization (WHO) references [31].

#### 2.3.2. Blood Pressure

Systolic and diastolic blood pressures (mmHg) were measured at 8 years of age using a Dinamap ProCare 100/200 digital blood pressure monitor following a standardized procedure. Blood pressure was measured after at least 15 min from arrival at the centre and after at least 5 min of rest. The measurement was taken in duplicate and on the left arm supported by a slightly elevated horizontal support (that is, close to the level of the heart). Both measurements were taken separated by at least 5 min, and the mean between them was used for statistical analysis. Systolic and diastolic blood pressure are presented as raw measurements (mmHg). Furthermore, they were standardized as percentiles for height and sex using the references from the American Academy of Pediatrics, 2017 [32].

#### 2.3.3. Blood Sample Parameters

At 8 years of age, a fasting venous blood sample was drawn. Total cholesterol, low-density lipoprotein cholesterol (LDL cholesterol), high-density lipoprotein cholesterol (HDL cholesterol), triglycerides, insulin and glucose were analysed. Total cholesterol (mg/dL), LDL cholesterol (mg/dL), HDL cholesterol (mg/dL), triglycerides (mg/dL), and glucose (mg/dL) were assessed in the clinical chemistry laboratories of study centres with routine methods used for clinical diagnosis [33]. Total and HDL cholesterol, triglycerides and glucose were analysed by indirect or enzymatic potentiometric methods. LDL cholesterol values were calculated by the Friedewald equation [34]. Insulin (AIU/mL) was quantified at the Department of Biochemistry, Radioimmunology and Experimental Medicine of the Institute for Children’s Memorial Health using the immunoradiometric assay (DiaSource, Nivelles, Bélgica) [35]. The evaluation of the homeostasis model for insulin resistance (HOMA-IR) was calculated as an approximation for insulin resistance [36,37].

#### 2.3.4. Cardiometabolic Risk Assessment

The cardiometabolic risk factor was calculated as a continuous variable equal to the sum of the internal z-scores for both sexes separately of the waist circumference, HDL cholesterol (multiplied by -1 to equate the sense of its beneficial effect to the other parameters), LDL cholesterol, triglycerides, HOMA-IR, DBP percentile and SBP percentile, as described in other studies [2,38,39]. A higher score was indicative of a less favourable cardiometabolic profile. This score could be summed to a maximum of around ±14 points.

### 2.4. Ethics

The study complied with the ethical requirements of the Declaration of Helsinki [40]. The study was approved by the ethics committees of all research centres and parents or legal guardians obtained the study information and signed the informed consent to participate.

### 2.5. Statistical Analyses

Continuous normally distributed variables are shown as mean and standard deviation, and continuous not normally distributed variables are presented as median and interquartile range. The distribution of the sample was assessed by means of graphical representation. Categorical variables are described as n and percentage of the total. Differences between sexes in continuous variables were assessed either through Student’s *t*-test or Mann-Whitney U-test according to the distribution of the variables. A cross-sectional analysis of fibre intake (g/1000 kcal) and its relationship with health variables at 8 years was performed using linear regression analysis, in which fibre intake (total or by subtypes) was adjusted by country of origin (Germany, Belgium, Italy, Poland and Spain), sex, average energy intake (kcal/day), maternal education (high, medium or low) and BMI (when this was not the dependent variable of the model). The feeding type (intervention low vs. high protein content during the first year) and breastfeeding (at least 3 months) was considered for adjustment in linear regression models but discarded after checking that there was no effect on the cardiovascular health related markers at that age.

The relationship between repeatedly staying in a low fibre intake tertile throughout childhood (at least 3 timepoints out of 5 between the 3 and 8 years of age) and health variables at 8 years was analysed using one-way ANOVA models adjusted for country of origin, maternal education, sex and BMI. For this analysis, participants who had reported fibre intakes in the lowest tertile on at least 3 occasions were considered.

Statistical significance was accepted at the level of *p* < 0.05.

All statistical analyses have been conducted with IBM SPSS Statistics for Windows, Version 26.0 Released 2019 (IBM Corp. Armonk, NY: IBM Corp).

## 3. Results

A total of 587 children attended the visit at age 8 years (277 boys, 310 girls). Blood pressure was obtained in 556 children (53.2% were girls). Among those, 400 participants (51.8% girls) completed the 3-day dietary diary. From these, 361 (49.9% girls) fasted blood samples were obtained.

### 3.1. Dietary Intake

Table 1 describes the daily intakes of energy, macronutrients and fibre by age. Stratification and comparison by sex is shown in Appendix A, Table A1.

Table 2 describes the total fibre intake and that adjusted for energy according to the food source and in all age groups. The sources of consumed dietary fibre were divided into four food groups based on the corresponding fibre type. Stratification and comparison by sex is shown in Appendix A, Table A2.

### 3.2. Association between Dietary Fibre Intake and Health in Children—Cross-Sectional Analyses

The cardiovascular health-related variables (anthropometric values, the biochemical parameters and blood pressure) at 8 years are described in Table 3. Linear regression analyses shows that children who ate a higher amount of fibre at 8 years of age had a higher BMI, but the model explains that only the 4.8% of total BMI at 8 years (Appendix B, Table A3). No further associations were found in cross-sectional analyses between fibre intake and health outcomes.

### 3.3. Fibre Intake According to Dietary Source and Association to Health—Cross-Sectional Analyses

When relating dietary fibre intake according to its source, a higher intake of soluble fibre (fibre from fruit and vegetables) at 8 years was related to lower HOMA-IR and cardiometabolic risk score (Table 4). Systolic blood pressure was inversely associated with the intake of fibre from nuts and legumes and was directly associated with the consumption of resistant starch. Triglycerides levels were not associated to dietary fibre intakes from different food sources.

### 3.4. Association between Dietary Fibre Intake and Health in Children—Longitudinal Analyses

Of the total of 400 children who attended the 8-year visit with dietary diary, 89 had repeatedly remained in the lowest tertile of dietary fibre intake between 3 and 8 years (at least three times out of five in that period). Children staying repeatedly in the lowest tertile of fibre intake throughout childhood, at 8 years, exhibited statistically significant higher glucose levels, higher HOMA-IR, higher cardiometabolic risk score and a trend for a higher systolic blood pressure, as shown by ANOVA adjusted models (Figure 1). Models carried out on triglycerides, HDL and LDL cholesterol and diastolic blood pressure did not reveal any significant association.

## 4. Discussion

This is the first multicentre European study investigating the association of dietary fibre intake and its sources with cardiovascular health in young children.

Fibre dietary intakes at 8 years old did not show any significant association to cardiovascular health at the same age. However, maintaining low fibre intakes during young childhood was associated with a worse glucose tolerance, systolic blood pressure and overall cardiometabolic risk at the age of 8. Not all fibre types were associated with these benefits. The consumption of fibre from fruits and vegetables was associated with a lower insulin resistance index and cardiometabolic risk, and the consumption of fibre from legumes and nuts was associated with a trend to lower SBP. These associations were not explained by differences in energy intake, as fibre intake was adjusted for total energy intake. Insoluble fibre intake was not associated to health outcome parameters and resistant starch was associated with a trend to a higher SBP.

Different prospective studies and clinical trials in adults have shown that following a diet rich in fibre is related to a lower cardiometabolic risk by improving the lipid profile, reducing blood pressure and insulin resistance [18,41,42,43]. Although the metabolic syndrome is well defined in adults, there is no universal and uniform definition for children and adolescents [44,45] and frequency in this age group is low. This was the reason why we used a continuous score as a cardiometabolic risk score, using common variables of the metabolic syndrome for the analyses of our study [38].

In children, studies that link dietary fibre intake and cardiometabolic health are scarce and mostly in adolescents or pre-adolescents [46,47,48,49,50]. We found only two published studies that analysed the relationship between dietary fibre intake and cardiometabolic risk in young children, which reported associations to lipid profiles but not to glucose metabolism [2,23].

In adults, effects of fibre intake on weight control and lower obesity risk have been attributed to different mechanisms, including the satiating power of fibres [10,51]. In children, concern was raised that a high fibre intake could decrease energy intake and the absorption of critical nutrients, thereby inducing potential harm. Data from the several publications have shown that high fibre intakes do not cause growth problems or lack of nutrients in healthy children [49,52]. In our study, the children who consumed more fibre, regardless of its source, had a higher BMI (without reaching pathological variations). This association was independent of the total energy intake, and the results obtained in our analyses associating dietary fibre consumption and cardiovascular health were independent of the BMI. Therefore, increasing the intake of dietary fibre can improve cardiovascular health risk markers in the short and long term, even in young children independently of weight.

Dietary fibre intake improves glucose homeostasis by causing a decrease in blood glucose, a reduction in hyperinsulinism and an improvement in insulin resistance [17,43]. This effect has been classically attributed to soluble fibre (which, through the formation of viscous gels in the intestine, reduces glucose absorption and decreases insulin secretion), but some studies have found significant effects from insoluble fibre as well [10,51,53]. In children, the relationship between fibre intake and glycaemic control had not been well demonstrated [24]. Our results are novel, as we have observed a similar trend to that from adults at an early age. Thus, a sustained low fibre intake from childhood could be associated to a poorer glucose tolerance. Furthermore, a diet with a higher amount of soluble fibre at the age of 8 from fruits and vegetables could reduce insulin resistance index as several studies in adults have previously shown [10,17,54].

Similar to studies previously published that have focused on children, adolescents and adults [23,49,50], we found a trend for a lower LDL cholesterol at 8 years of age with higher fibre intakes. The improved lipid profile may occur through several mechanisms, such as the reduction of cholesterol absorption, the blocking of its hepatic synthesis (through propionate derived from fibre fermentation) and the reduction of circulating cholesterol (due to increased cholesterol utilisation for the synthesis of bile acids, stimulated by intestinal bile acid sequestration by fibre) [55].

Higher fibre intakes have been related to lower systolic and diastolic BP in adults, as well as an improvement in BP levels in hypertensive patients [56,57,58]. In children, studies linking dietary fibre intake and BP are very scarce. Our study corroborates the association found in adults for the first time in children. Children with low fibre intakes maintained over time exhibited higher figures of systolic blood pressure. One explanation that could partly explain this association is that fibre-rich vegetables could be poorer in sodium and rich in potassium (depending on its preparation). In any case, a mechanism that could explain this association between fibre and blood pressure could be the renal sodium absorption, increased activity of the sympathetic nervous system, and sensitivity induced by free fatty acids to adrenergic stimuli and the antagonized vasorelaxation of nitric oxide that produces hyperinsulinism [59]. In line with previous studies, this protective association of fibre with blood pressure was mainly related with soluble fibre and that derived from pulses and nuts [58,60,61,62,63,64,65]. Nuts also contain high levels of unsaturated fatty acids and bioactive compounds that could contribute to the improvement of blood pressure and endothelial function [66,67]. On the other hand, pulses have a high content of dietary fibre, vegetable protein and potassium, which confer reducing effects on blood pressure [68]. The intake of resistant starch was associated with higher SBP. Resistant starch mainly comes from potatoes, and potatoes consumption, particularly as fries or chips, could be frequently accompanied by a high salt intake.

The amount of salt could not be quantified, which is considered a limitation in this study. Other dietary factors that could be related to cardiovascular risk, such as the proportion of sugars and fats, were not included in these analyses. However, all of the analyses were adjusted by overall energy intake. Another possible limitation is that the analyses did not account for physical activity, which might be related to the health outcomes as well. Other possible limitations of the study are the decreasing number of participants with increasing age, especially those who provided all dietary data. A strength of the study is that this is the only longitudinal European multicentre study in prepubertal children in which fibre and its food sources are analysed in relation with different cardiovascular health items and cardiometabolic risk.

## 5. Conclusions

In summary, children who consistently consumed fewer amounts of fibre had indications of poorer glucose tolerance and overall higher cardiometabolic risk score at 8 years. These better glucose tolerance levels and overall higher cardiometabolic risk scores were mainly associated with a higher intake of fibre from fruits and vegetables. These results support the importance of acquiring a healthy diet from childhood that includes dietary fibre as an important component. A diet rich in dietary fibre from fruits, vegetables, legumes and nuts should be promoted from an early age, regardless of the weight and health status of children, due to its apparent short- and long-term benefits.

## Figures and Tables

**Figure 1 nutrients-13-00012-f001:**
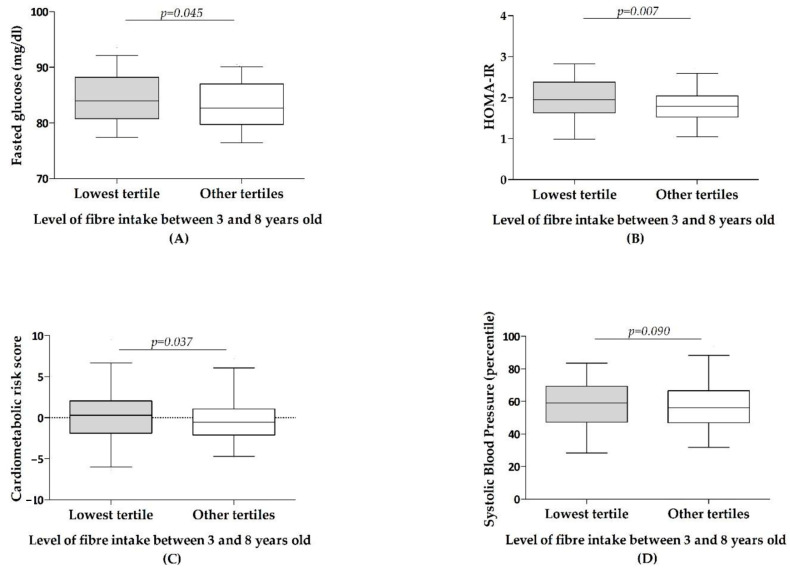
Cardiovascular health parameters according to the level of fibre intake throughout childhood. Children classified as being repeatedly in the lowest tertile of fibre intake between 3 and 8 years old (at least three times out of five in this period) or in other tertiles and/or not repeatedly in the lowest tertile. Results from one-way ANOVA models adjusted by country, sex, body mass index and maternal education for glucose (**A**), HOMA-IR (**B**), Cardiometabolic risk (**C**) and systolic blood pressure (**D**).

**Table 1 nutrients-13-00012-t001:** Energy, macronutrient and fibre intakes by age group.

	Energy (kcal/Day) Mean (±SD)	Proteins (g/Day) Mean (±SD)	Lipids (g/Day) Mean (±SD)	Carbohydrates (g/Day) Mean (±SD)	Total Fibre (g/Day) Mean (±SD)	Total Fibre (g/1000 kcal) Mean (±SD)
3 years, *n* = 534	1221 (243)	47.0 (12.2)	47.1 (12.8)	152.6 (37.2)	8.3 (3.3)	6.9 (2.7)
4 years, *n* = 504	1317 (249)	50.0 (13.1)	52.0 (14.4)	163.2 (34.5)	9.2 (3.4)	7.1 (2.4)
5 years, *n* = 447	1394 (268)	52.1 (13.6)	54.7 (14.4)	174.5 (41.1)	9.9 (3.5)	7.2 (2.2)
6 years, *n* = 469	1479 (253)	55.2 (12.7)	58.1 (14.3)	184.8 (39.5)	10.8 (3.6)	7.3 (2.4)
8 years, *n* = 400	1598 (304)	61.1 (15.4)	65.4 (17.8)	192.4 (42.1)	12.1 (4.0)	7.7 (2.3)

**Table 2 nutrients-13-00012-t002:** Fibre intake by age according to food source.

	Insoluble		Soluble		Resistant Starch		Fibre from Pulses and Nuts	
	g/Day	g/1000 kcal	g/Day	g/1000 kcal	g/Day	g/1000 kcal	g/Day	g/1000 kcal
3 years, *n* = 531	3.32 (2.32, 4.41)	2.73 (1.99, 3.62)	3.27 (1.99, 4.98)	2.72 (1.71, 4.08)	0.36 (0.10, 0.66)	0.30 (0.09, 0.54)	0.00 (0.00, 0.00)	0.00 (0.00, 0.00)
4 years, *n* = 503	3.68 (2.63, 4.94)	2.86 (2.15, 3.69)	3.80 (2.41, 5.24)	2.87 (1.79, 3.89)	0.39 (0.07, 0.73)	0.30 (0.05, 0.53)	0.00 (0.00, 0.01)	0.00 (0.00, 0.00)
5 years, *n* = 445	4.15 (3.21, 5.38)	3.06 (2.41, 3.73)	3.76 (2.43, 5.47)	2.65 (1.77, 3.84)	0.39 (0.07, 0.73)	0.27 (0.05, 0.52)	0.00 (0.00, 0.01)	0.00 (0.00, 0.01)
6 years, *n* = 468	4.78 (3.64, 5.85)	3.17 (2.57, 3.94)	3.85 (2.46, 5.93)	2.65 (1.73, 3.85)	0.41 (0.10, 0.87)	0.30 (0.07, 0.59)	0.00 (0.00, 0.01)	0.00 (0.00, 0.00)
8 years, *n* = 399	5.68 (4.53, 7.23)	3.60 (2.94, 4.49)	4.03 (2.61, 5.80)	2.58 (1.72, 3.65)	0.55 (0.18, 1.02)	0.35 (0.11, 0.63)	0.00 (0.00, 0.02)	0.00 (0.00, 0.01)

Data presented as median (IQR). Insoluble fibre includes fibre from cereals, pasta, rice, cookies, cakes and convenience foods. Soluble fibre includes that from fruits and vegetables. Resistant starch includes fibre from potatoes and potato products. Pulses and nuts are groups separately, containing soluble, insoluble and resistant starch.

**Table 3 nutrients-13-00012-t003:** Cardiovascular health-related parameters at age 8 years.

	All	Boys	Girls
**Anthropometry**	**Mean (±SD)**	**Mean (±SD)**	**Mean (±SD)**
Weight (kg)	28.6 (6.1)	28.7 (6.6)	28.6 (5.7)
Height (cm)	129.6 (5.7)	130.2 (5.8)	129.0 (5.6) *
BMI (kg/m^2^)	16.9 (2.6)	16.8 (2.8)	17.0 (2.4)
Abdominal circumference (cm)	59.4 (7.3)	59.5 (7.5)	59.3 (7.1)
Weight for age (z score)	0.59 (1.19)	0.59 (1.31)	0.60 (1.07)
Height for age (z score)	0.43 (0.99)	0.47 (1.03)	0.39 (0.96)
BMI for age (z score)	0.46 (1.21)	0.40 (1.37)	0.52 (1.04)
**Biochemical Parameters**	**Mean (±SD)**	**Mean (±SD)**	**Mean (±SD)**
Glucose (mg/dL)	84 (8)	84 (7)	83 (8) *
Total cholesterol (mg/dL)	167 (27)	164 (27)	169 (27)
HDL cholesterol (mg/dL)	60 (15)	61 (16)	59 (14)
LDL cholesterol (mg/dL)	94 (25)	91 (24)	97 (25) *
Triglycerides (mg/dL)	60 (26)	55 (22)	64 (29) ^ǂ^
Insulin (µIU/mL)	8.75 (3.15)	8.43 (2.96)	9.09 (3.32) *
HOMA-IR ^§^	1.82 (0.70)	1.78 (0.66)	1.88 (0.74)
**Blood Pressure**	**Mean (±SD)**	**Mean (±SD)**	**Mean (±SD)**
Systolic blood pressure (mmHg)	100 (10)	100 (9)	100 (10)
Diastolic blood pressure (mmHg)	57 (7)	56 (7)	58 (7) *
Systolic blood pressure (percentile)	57.8 (27.5)	55.6 (27.9)	59.7 (28.2)
Diastolic blood pressure (percentile)	44.5 (21.8)	41.5 (21.4)	47.2 (21.9) *
**Cardiovascular Risk**	**Mean (±SD)**		
Cardiometabolic score	−0.20 (3.85)	−0.03 (3.87)	−0.39 (3.86)

* *p* < 0.05, ^ǂ^
*p* < 0.01, *p*-value for Student’s *t*-test for comparison boys vs. girls. Skewed variables tested in logarithmic form; ^§^ Not normally distributed variable, median 1.70 (IQR: 1.31, 2.19); boys median 1.66 (IQR: 1.32, 2.10); girls median 1.78 (IQR: 1.31, 2.25).

**Table 4 nutrients-13-00012-t004:** Intake of dietary fibre from different sources and cardiovascular health parameters at age 8 y.

		HOMA-IR			Systolic Blood Pressure			Triglycerides			Cardiometabolic Risk Score		
		B	95% CI (Min, Max)	*p*-Value	B	95% CI (Min, Max)	*p*-Value	B	95% CI (Min, Max)	*p*-Value	B	95% CI (Min, Max)	*p*-Value
Unadjusted analyses	Insoluble fibre	0.005	(−0.008, 0.017)	0.445	1.351	(0.13, 2.56)	0.030	0.006	(−0.006, 0.017)	0.346	0.148	(−0.130, 0.426)	0.296
	Resistant starch	0.028	(−0.009, 0.065)	0.137	9.698	(4.67, 14.72)	<0.001	0.015	(−0.020, 0.050)	0.397	1.323	(0.495, 2,151)	0.002
	Soluble fibre	−0.010	(−0.018, −0.001)	0.028	−0.358	(−1.41, 0.69)	0.505	−0.001	(−0.009, 0.006)	0.709	−0.068	(−0.261, 0.124)	0.486
	Fibre from pulses and nuts	0.015	(−0.004, 0.034)	0.131	−1.917	(−4.49, 0.65)	0.144	−0.008	(−0.026, 0.010)	0.381	−0.301	(−0.734, 0.131)	0.171
		R^2^ = 0.016			R^2^ = 0.040			NS			R^2^ = 0.37		
Adjusted analyses	Insoluble fibre	0.000048	(−0.010, 0.010)	0.982	−0.48	(−1.70, 0.74)	0.440	0.006	(−0.005, 0.016)	0.281	0.059	(−0.14, 0.26)	0.566
	Resistant starch	−0.014	(−0.046, 0.017)	0.370	4.60	(0.06, 9.19)	<0.0496	−0.023	(−0.057, 0.011)	0.104	0.087	(−0.56, 0.74)	0.793
	Soluble fibre	−0.008	(−0.016,−0.001)	0.025	−0.19	(−1.16, 0.77)	0.694	−0.005	(−0.014, 0.001)	0.182	−0.159	(−0.30, −0.00)	0.037
	Fibre from pulses and nuts	0.012	(−0.004,0.028)	0.148	−2.24	(−4.53, 0.05)	0.055	0.012	(−0.007, 0.027)	0.183	−0.045	(−0.37, 0.20)	0.784
		R^2^ = 0.346			R^2^ = 0.252			R^2^ = 0.207			R^2^ = 0.489		

Cross-sectional analyses on the intakes of different fibre types adjusted by energy (g/1000 kcal) at 8 years old on health outcome parameters at the same age. All fibre types entered together in a linear regression model, unadjusted and adjusted by country, sex, average energy intake at 8 years old, maternal education and BMI. NS: Overall goodness of fit of the model was not significant.

## Data Availability

Data available on request due to ethical restrictions.

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
