# Peer review of "Fibre Intake Is Associated with Cardiovascular Health in European Children"

_nutrients, 2020, doi:10.3390/nu13010012_

Round 1
Reviewer 1 Report
The authors seek to "analyse the relation of dietary fibre intake in children from fiver European countries at the ages of 3 to 8 years to cardiovascular health markers such as obesity, blood lipids, blood pressure and glucose metabolism." The data appear unique and collected using rigorous methods to produce reliable results. The authors identify the novelty of their data compared with other datasets to explore the proposed relationship, despite seemingly being collected in what seems to be only as recent as 2012-2013. The strengths of this paper include the longitudinal dietary data among children 3 to 8 years old from diverse settings. However, the authors tend to use causal language throughout the paper, which should be changed to associative language based on the study design. The paper can be further improved through more robust statistical analyses, including sensitivity analyses that can help tests the reliability of the results. I also think the results, at least the descriptive data, could be presented more clearly, including stratification by sex. The specific comments are designed to strengthen the paper.
TITLE/ABSTRACT
- Title: Change "effects" to "association" here and throughout the paper with other language such as "increased" rather than "higher" as another example (Lines 36, 172, 196, 197, 214, 284).
- Abstract: Sample description, even if brief, is needed, such as proportion female.
MATERIALS AND METHODS
- Section 2.1: Were more updated data available through this study? If so, then this would strengthen the paper further.
- Section 2.3: Were risk factor data collected prior to 8 years old? It may be helpful to incorporate these time-varying covariates into the paper's statistical models, though there may be a risk of overfitting.
- Section 2.3/Line 131: Readers will need more specific details about the cardiovascular risk score, which will not be familiar to many, if not most. At a minimum, the range of the scale needs to be added here and in Table 3.
- Section 2.5: Some data appear skewed in Table 2 and median (IQR) are used on occasion, but distribution of data can be rechecked throughout with updated methods to account for these analyses.
- Section 2.5: How were missing data handled? A flowchart may help to support the text in Lines 152-153 as a supplement. Sensitivity analyses using multiple imputation would strengthen the paper and its results.
- Section 2.5: What threshold for statistical significance was used? Was there any correction for multiple testing?
- Section 2.5: I think the use of the ANOVA may not be as robust as a generalized linear mixed model to account for time varying covariates. I also favor using continuous covariates, rather than tertiles as reported in Figure 1.
RESULTS
- Tables 1 and 2: The orientation should be changed, and I would compare either across age, sex, or both. P values are needed to compare across the groups.
- Table 3: Stratification by sex would be helpful here. Define cardiometabolic score so the table can stand on its own.
- Table 4: This seems like it could moved to the supplement.
- Figure 1: I would avoid using tertiles altogether but would at least report results by all tertiles separately. I favor reporting of unadjusted and adjusted analyses so readers can understand the attenuation
- Table 5: Were these results based on data collected throughout the study period? This could be clearer in the legend or footnotes of this table. Favor reporting unadjusted and adjusted analyses.
DISCUSSION
- Lines 203-205: Delete
- Lines 206: This seems like an atypical summary of the key results, especially when there is evidence that fiber intake may increase over time but I would contend that this could be driven, at least in part, because of differential missingness by baseline fiber consumption. I would expect individuals with higher fiber intake, who were generally healthier, to be more likely to follow-up.
- Line 224: I would de-emphasize metabolic syndrome here and throughout. There appears less and less evidence to support this anything more than the sum of the individual risk factors rather than being synergistic as initially hypothesized.
- Line 234: I did not understand the use of "expense" here, which has a more negative connotation.
- Overall, I would have preferred a discussion that sought to fill the gap in recommending absolute fibre intake recommendations as outlined in the Introduction. The discussion is long at present and the discussion of the relationship between fibre and blood pressure in adults seems to minimize potential confounding driven by higher fibre foods being higher in potassium and lower in sodium based on the mode of preparation. The latter point is briefly alluded to, but these are likely more important factors for blood pressure than fibre alone. I recognize that disentangling these relationships is challenging
Author Response
We gratefully acknowledge the deep and consistent review, as well as the dedicated care to all details that will for sure improve the manuscript. We have addressed all the comments made by Reviewer 1 as far as possible.
The reviewer will see, that the lines he/she was giving as reference for corrections do not match at all with the lines I refer to. It seems to me that the “view” of the reviewer is completely different from that of the author. We hope the changes can still be easily checked though.
Please, see the attachment
Kind Regards
Luque and Escribano

Reviewer 2 Report
This is an interesting study where the authors assess dietary fibre intake and cardiovascular health markers in a sample of children. The authors compare dietary fibre intake (and type of dietary fibre intake) at 5-year intervals. During the 5th year (aged 8 years), cardiovascular health markers were also measured. A cross-sectional analysis at year 5 demonstrated a positive association between dietary fibre intake and higher BMI – no other association with any cardiovascular health marker was observed.
However, in further analyses, children who had consistently remained in the lowest tertile for dietary fibre intake over the 5-year study period were found to have significantly higher fasting glucose, insulin resistance, systolic BP levels and a higher cardiometabolic risk score compared to subjects in other tertiles. With regard to specific fibre intake, in further analyses, a higher intake of soluble fibre was found to be related to insulin resistance and the cardiometabolic risk score.
Although the sample size is modest (n=587), and is further reduced according to analyses undertaken – the paper is well-written, the data are displayed well and the analyses undertaken are appropriate for the study.
I have a few comments which the authors may wish to consider:
Line 184: “Of the total of 538 children who attended the 8-year visit”. This should be re-worded, as it was the 5th visit by children at 8 years of age. Also – previously in the manuscript, the authors state that food dietary diaries were only available for 400 children during the 5th visit. So, surely this analysis is restricted to 400 children? In other words – the data presented in Figure 1 should only include children for whom dietary data were available during each 5-year period. This is not clear.
Line 194: “When relating dietary fibre intake according to its source..” This looks like a cross-sectional analysis using data from the 5th year collection period (8 years of age) – in which case in should be presented after Table 4 and before Figure 1.
Line 203: First paragraph. These are the “Nutrients” journal guideline instructions – and should be omitted from the manuscript.
Line 206: “This is the first multicentre European longitudinal study” – although data were collected over a 5-year period – many of the analyses conducted in this study are not prospective – so this should be re-worded. Also, the sentence “The results show, that dietary fibre intake adjusted by energy, remains steady during 5 years, highlighting the importance of early education to develop healthy dietary habits”. It is not clear to me what that means.
Line 236: “A basic component of the metabolic syndrome is BMI and its relationship with the development of overweight and obesity”. BMI is a measure of overweight and obesity. Also – a basic component of metabolic syndrome is usually thought to be abdominal obesity measured through waist circumference. I would restructure the sentence to read “A basic component of the metabolic syndrome is overweight and obesity…”.
Line 246: “Therefore, increasing the intake of dietary fibre can improve cardiovascular health risk markers in the short and long term, even in young children with normal weight.” – I’m not sure the findings support that statement. The authors did not classify subjects as normal weight, overweight or obese.
Line 280: The authors state that the amount of dietary salt intake could not be quantified, which is a limitation of the study as this could not be adjusted for in analyses. There are other dietary intakes which were also not accounted for in analyses – such as sugar intake and fats. This should be discussed.
There are a few typos in the manuscript (which is normal) so the paper could use a further proof read.
Author Response
We gratefully acknowledge the deep and consistent review, as well as the dedicated care to all details that will for sure improve the manuscript. We have addressed all the comments made by Reviewer 1 as far as possible.
The reviewer will see, that the lines he/she was giving as reference for corrections do not match at all with the lines I refer to. It seems to me that the “view” of the reviewer is completely different from that of the author. We hope the changes can still be easily checked though.
Please, see the attachment
Luque and Escribano

Reviewer 3 Report
In this manuscript, Larrosa et al. present their observations related to fibre intake in children and cardiovascular health at 8 years of age. Data are derived from a double-blinded, randomized dietary intervention. However, the current analyses are apparently not related to the intervention per se.
Major comments
My main concern is the intervention itself. In the manuscript, the authors do not give any details of the dietary intervention. Importantly, based on the manuscript, it cannot be judged whether the intervention could have had some impact on the cardiovascular health markers investigated in the study.
Minor comments.
The text needs language revision. Some language problems are highlighted below, but these are far from all that should be addressed.
Row 25. …between dietary fibre during childhood… -> …between dietary fibre intake during childhood… This similar mistake is repeated in many other parts of the text.
Rows 51-52. How is it useful that insoluble fibre hinders the absorption of nutrients?
Give abbreviations only to those terms that you use later in the text. For example, row 54, you do not use SCFA later in the text, so there is no need to abbreviate it. On the other hand, if you give abbreviation (as for TG in row 121), you should use it in the subsequent text (see rows 122 and 124, for example).
Rows 66-67. “…because of a lack of data the effects of dietary intake in childhood on long-term health outcomes.” Please revise.
Row 82. The word “data” is plural. Therefore: “The data used for the study were…”
Why was energy intake presented using calories, and not SI unit?
Row 138. The ethics committees are usually called “ethics committees”, not “ethical committees”. After all, we cannot say if they are ethical or not, but what we do know is that they deal with the matters of ethics.
Row 147 onwards. Here you talk about “repeatedly staying in a low fibre intake tertile”, but later you talk about “at least 3 occasions”. These do not mean the same thing. Please revise the text to be consistent.
Gender is a cultural term (gender studies, gender identity). Sex is a biological term (girls, boys). I think the correct term to use in your manuscript is “sex“.
In the results, you say that energy, macronutrient, and fibre intakes (as also presented in Table 1) increased with age. However, you provide no p-values to support this statement. Similarly, for the results presented in Table 2 and the related text. If you state that increase took place, this needs to be supported by the p-value.
Table 2. Why do you have a lower number of participants here compared to Table 1?
Row 172. What do you mean by saying that “children who ate a higher proportion of fibre”? How did you measure the proportion of fibre eaten?
Throughout the results, you suggest that fibre intake “increased” something. For example row 172: “…children…had an increased BMI…” How did you measure this increase? I could not find any results stating the amount of increase. For what I understand, you should be talking about “higher BMI”, not “increased BMI”. Also see rows 196-197 (reduction/increased) and row 287 (improvements in insulin resistance…). There may be other such statements, too, so please go through the entire paper.
Table 3. Please check the unit of insulin (µUI/ml or µIU/ml)?
Table 3. The results for Diastolic blood pressure are given twice.
Row 184. Here you say that there were a total of 538 children attending the 8-year visit. However, on row 152 you talk about 587 children.
Figure 1. Why were other analyses adjusted for country, sex, BMI, energy intake, and maternal education, but these analyses did not include energy intake, and maternal education?
Rows 196-197. The results dealing with SBP were not statistically significant. Please modify the text accordingly.
Rows 203-205. Please remove the text.
Row 244. Calorie is a unit of energy. Thus, the actual phenomenon is ENERGY! Please do not use a term “caloric intake”, but “energy intake”, instead.
Row 287. In the results, fibre intake was not associated with LDL-cholesterol concentration. Please amend the conclusions to be in line with the results.
Author Response
We gratefully acknowledge the deep and consistent review, as well as the dedicated care to all details that will for sure improve the manuscript. We have addressed all the comments made by Reviewer 1 as far as possible.
The reviewer will see, that the lines he/she was giving as reference for corrections do not match at all with the lines I refer to. It seems to me that the “view” of the reviewer is completely different from that of the author. We hope the changes can still be easily checked though.
Please see the attachment
Luque and Escribano

Round 2
Reviewer 1 Report
The authors have been generally responsive to my prior comments.
Author Response
As there were no specific requirements from Reviewer 1, we herewith just report that we have done improvements in language and spelling mistakes.
Thanks for the suport for revieweing the manuscript
Reviewer 3 Report
Major comments
The original intervention study should be better described. Indeed, when describing the study design, the authors should clearly state that this is a secondary analysis of data collected in an intervention study. After stating this, this intervention should be described in sufficient detail.
The outcomes, investigated, are also related to physical activity, which was not taken into account. This should be stated as a limitation.
Minor comments.
The text still needs language revision. Some points are addressed below, but it is not possible for the reviewer to point out all that should be addressed.
In general, the numerical value and the following unit should have a gap in between. Thus, for example, 10g precision -> 10 g precision. Also there should not be a gap in kg/ m2 (and “2” should be written superscript).
Abstract. Results: four-hundred -> Results: Four-hundred.
The terms “females” and “males” are suitable for describing animals. Humans should be called “women” and “men”. Or in this case “girls” and “boys”.
The list of keywords. You mention both “dietary fibre” and “fibre”. One should be sufficient.
2.2 Dietary intake assessment. “The average intake of energy, fibre, and macronutrients…were…“ -> “The average intakes of energy, fibre, and macronutrients…were…”
2.3 Health variables. “Anthropometry” is the science of obtaining systematic measurements of the human body. Therefore you cannot say that “anthropometry was a main health outcome”.
Blood pressure. You state that the two BP measurements were separated by a mean time of 5 minutes. So some of the measurements were conducted with less than 5 minute´s interval? What was the range of these intervals?
Blood sample parameters. For the most, you write “LDL cholesterol” but in one sentence you use “LDL-C”. Please be consistent.
Cardiometabolic risk assessment. “The cardiometabolic risk factor was calculated…..for each sex of the waist circumference…” -> “For both sexes separately, the cardiometabolic risk factor was calculated…”
Cardiometabolic risk assessment. “…this score could sum up a maximum of…” -> “…this score could sum up to a maximum of…”
Statistical analyses. In the first sentence, you describe how you present the continuous variables as mean and SD, but in the next sentence you say that non-normally distributed variables are actually presented as median and IQR. Please be consistent.
2.5 Statistical analyses. The relationship between repeatedly staying in a low fibre intake…. This sentence is missing the maternal education as one of the confounders.
Results. First paragraph, second sentence. Please remove “most of them”.
3.1. Dietary intake. You state here that energy, macronutrients [SIC], and fibre intakes increased with age. Please provide a p-value to back up this increase. Also: “fibre (intake) adjusted by (for?) energy increased (not raised!) with age” -> provide a p-value for this statement. Finally: “…fibre (intake) from pulses and nuts increased…” -> provide a p-value.
3.2 Association between dietary fibre intake and health in children – cross-sectional analyses. Here you state that these are cross-sectional analyses, but in the next title (3.3 Fibre intake according to dietary source and association to health) you do not state this. I believe, however, that also these analyses were cross-sectional. I suggest you structure the results in a way that better enables the reader to see in a glance which are results from cross-sectional analyses and which are results from longitudinal analyses.
Table 3. Whole –> All. For “anthropometry”, boys do not have standard deviation after the “mean”. Also “Mean (±SD)” is missing from Boys and Girls for Biochemical parameters onwards. For SBP and DBP it is not clear why the results are shown twice. You need to provide explanation why they are different. For variables that are not normally distributed, the results should be given as median (IQR), as you state in the methods. You provide an asterisk for height for girls, and explain it in the footnote. However, you give the same asterisk for p<0.05, too. Use only one asterisk with one explanation. In the footnote, you mention Student’s t-test, however, this was not described in the methods section. Please ensure that all methods are described accordingly.
Table 4. Why the R squared is not provided for triglycerides (unadjusted)? Please ensure that you are actually giving min and max values with B and p-value. I was wondering, if they are actually 95% confidence intervals, instead? If they are truly min max values (and even if they are “only” 95% CI), based on the adjusted SBP results for resistant starch (0.06 – 9.19), I would guess that the p-value should be <0.050 and not 0.050. Indeed, in the methods you state that p-values <0.050 denote statistical significance, and in the text you discuss that this SBP observation WAS significant. If these are correct, I suggest you change the p-value 0.050 -> <0.050 (or the actual p-value, for example 0.049).
3.4 Association between… There is repetition in the first two sentences.
Figure 1. Based on the submitted material, it was not possible to say which of the figures were final and which were supposed to be deleted. I assumed that only 4 small figures were meant to be “final”. The lower four looked better. However, I do not think the text “Fibre intake between 3 and 8 years of age repeatedly in the lowest tertile” goes well with the “lower tertile” and “not lower tertile” categories. Please re-consider the terms used.
Discussion. “The consumption of fibre from fruits and vegetables was associated with a reduction…” No reduction took place. Please rephrase.
Discussion. “A basic component of the metabolic syndrome is BMI and its relationship with overweight and obesity.” This sentence does not make any sense.
Discussion. SB -> SBP
Discussion, summary. “Improvements in glucose tolerance…” No improvements took place. Please rephrase.
Table A1. Please place the footnote at the end of the table, not in the middle.
Table A2. Title is missing girls. Also the table is missing explanation how the data are presented (Maybe median, IQR?).
Author Response
We gratefully acknowledge the detailed report from reviewer 2 and have addressed all of the reccomendations.
Find enclosed a detailed point by point letter.
